# A compact QCL spectrometer for mobile, high-precision methane sensing aboard drones

Béla Tuzson[1], Manuel Graf[1], Jonas Ravelid[1], Philipp Scheidegger[1], André Kupferschmid[2], Herbert Looser[1], Randulph Paulo Morales[1], and Lukas Emmenegger[1]

[1]Laboratory for Air Pollution / Environmental Technology, Empa - Swiss Federal Laboratory for Materials Science and Technology, Überlandstrasse 129, Dübendorf 8600, Switzerland
[2]Transport at Nanoscale Interfaces, Empa - Swiss Federal Laboratory for Materials Science and Technology, Überlandstrasse 129, Dübendorf 8600, Switzerland

**Correspondence:** Béla Tuzson (bela.tuzson@empa.ch)

**Abstract.** A compact and lightweight mid-IR laser absorption spectrometer has been developed as mobile sensing platform for high-precision atmospheric methane measurements aboard small, unmanned aerial vehicles (UAVs). The instrument leverages on two recent innovations: a novel segmented circular multipass cell (SC-MPC) design and a power efficient, low-noise, intermittent continuous-wave (icw) laser driving approach. A system-on-chip hardware control and data acquisition system enables energy-efficient and fully autonomous operation. The integrated spectrometer weighs $2.1\,\mathrm{kg}$ (including battery) and consumes $18\,\mathrm{W}$ of electrical power, making it ideally suited for airborne monitoring applications. Under stable laboratory conditions, the device achieves a precision ($1\sigma$) of $1.1\,\mathrm{ppb}$ within $1\,\mathrm{s}$ and $0.1\,\mathrm{ppb}$ $CH_4$ at $100\,\mathrm{s}$ averaging time. Detailed investigations were performed to identify and quantify the effects of various environmental factors, such as sudden changes in pressure, temperature, and mechanical vibrations, which commonly influence UAV mounted sensors. The instrument was also deployed in two feasibility field-studies: an artificial methane release experiment as well as a study on vertical profiles in the planetary boundary layer. In both cases, the spectrometer demonstrated its airborne capability of capturing subtle and/or sudden changes in atmospheric $CH_4$ mole fractions and providing real-time data at $1\,\mathrm{s}$ time resolution.

## 1 Introduction

Global emissions of methane are continuing to increase, making $CH_4$ a relevant component for managing realistic pathways to mitigate climate change, especially with near-term benefits (Shindell et al., 2012). However, the accurate quantification of the $CH_4$ budget and its variations still remain challenging, mainly due to uncertainties in $CH_4$ emissions from natural wetlands and fresh waters, and not sufficiently constrained partitioning of $CH_4$ emissions by region and processes (Kirschke et al.; Saunois et al., 2016). Therefore, there is a continuing need for technologies that enable the tracking and identification of methane sources at local scale and on a regular basis. In this context, lightweight and high-precision trace gas sensors aboard small unmanned aerial vehicles (UAVs) offer new opportunities for in-situ air pollution and emission monitoring (Golston et al., 2017). Combined with fast response, they can provide valuable information about the spatial and temporal variability of emissions regardless of terrain complexity or pollution source characteristics. This significantly extends the scale and the

level of detail of measurements compared to those provided by traditional stationary monitoring networks. The potential of this approach is well reflected by the increasing number of reported commercial as well as research-grade sensors and related applications (e.g. Berman et al., 2012; Villa et al., 2016, and references therein). The majority of these solutions are triggered by oil- and gas industry fugitive emission monitoring (Nathan et al., 2015; Yang et al., 2018; Golston et al., 2018; Fox et al., 2019; Martinez et al., 2020) and, correspondingly, are targeting elevated $CH_4$ concentrations near emission sources. It is, however, more challenging to develop a sensor for the investigation of natural emissions (e.g. O'Connor et al., 2010; DelSontro et al., 2010), which are typically more diffuse and much weaker. Here, we target these more demanding, but equally important application areas. Our aim is to drastically reduce the size of conventional lab-based laser spectrometers to an almost hand-held device, while maintaining the high precision required for atmospheric methane concentration monitoring. In the following sections, we describe our approach, present the hardware solution, and show the capabilities of the developed $CH_4$ sensor.

## 2 Methodology

### 2.1 Spectral selection

For high sensitivity methane detection using infrared absorption spectroscopy, there are mainly two suitable spectral regions corresponding to the strong fundamental absorption bands of C–H stretching ($\nu_3$ at $3.3\,\mu m$) and bending ($\nu_4$ at $7.7\,\mu m$) modes. Although, the stretching mode contains transition lines of up to five times higher intensity compared to the bending band, we still opted for the latter spectral range. This was motivated by two practical factors: i) distributed feedback quantum cascade lasers (DFB-QCL) with high optical power can readily be used, and ii) the $P$-branch of the $\nu_4$-band is at the edge of the atmospheric window, i.e. interfering absorption due to water vapor is considerably weaker. This is particularly important in the context of an open-path configuration, i.e. measurements at ambient pressure. Additionally, it is worthwhile to mention that the symmetric stretching vibration band ($\nu_1$) of nitrous oxide ($N_2O$) is also located in the vicinity and may be used at a later stage as a secondary tracer. Figure 1 shows the simulated transmittance of ambient air at atmospheric pressure for the spectral range covering the $P$- and $Q$-branch of the $\nu_4$-band of $CH_4$. It might be tempting to select one of the strong absorption feature resulting from many overlapping transitions in the $Q$-branch at $7.6\,\mu m$. However, given the narrow tuning range of a DFB laser source, this would make it very difficult to define the intensity baseline for a reliable spectral fitting. This is a consequence of the open-path configuration and the inherent atmospheric pressure broadening effect, which leads to large absorption features that eventually exceed the tuning range of the laser. In fact, the uncertainty of the baseline retrieval in the open-path system limits the performance in the $Q$-branch, especially when considering other influencing factors such as interferences from pressure-broadened, overlapping absorption features of other atmospheric species and changes in ambient temperature and pressure.

A systematic survey of the $7.7\,\mu m$ spectral region revealed an attractive window in the vicinity of a water absorption line at $1276.62\,cm^{-1}$ that exhibits a distinctly narrow profile even at ambient pressure (see inset Fig. 1). According to the HITRAN2016 database (Gordon et al., 2017), this absorption is given by the completely overlapping high-$J$ transitions, 15 0 15 $\leftarrow$ 16 1 16 and 15 1 15 $\leftarrow$ 16 0 16. The air-broadening coefficient of these transitions is $0.0064\,cm^{-1}\,atm^{-1}$, which is about

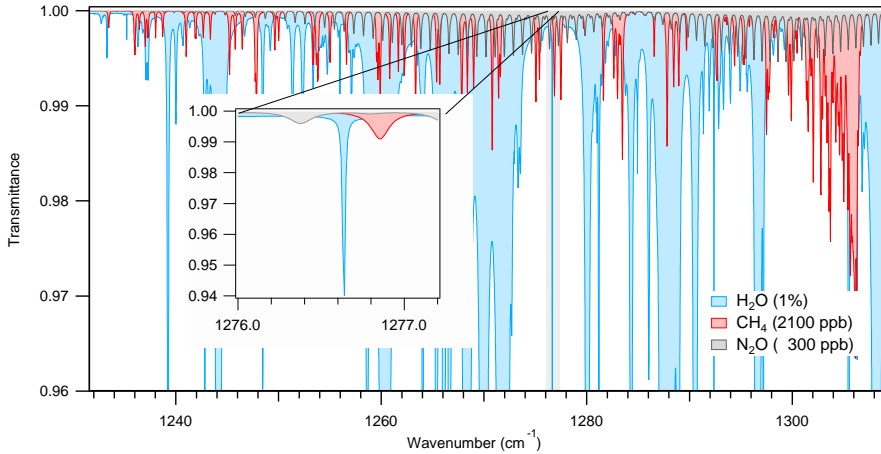

**Figure 1.** Simulated transmission spectrum of ambient air at atmospheric pressure over an optical path length of $10\,\mathrm{m}$. Only the contributing molecular species, e.g. $H_2O$, $CH_4$ and $N_2O$, with their mean atmospheric abundance are considered. The inset shows the selected spectral window for methane measurements. The water absorption line is markedly narrow, due to a particularly small pressure broadening coefficient of this transition.

ten times smaller than for transitions of low-$J$ states. This can be explained by inefficient collisional relaxation due to the wide energy separation ($\sim 300\,\mathrm{cm}^{-1}$) for high-$J$ states of $H_2O$ (Eng et al., 1972; Giesen et al., 1992). Considering the typical energies of translation motion ($kT \approx 200\,\mathrm{cm}^{-1}$), it is obviously difficult to take up the transition energies from such rotational states. Therefore, the perturbation of these high rotational energy states by collision remains low, which results in low pressure

broadening, so that the $H_2O$ absorption line at ambient pressure appears as narrow as it would be at $0.1\,\mathrm{atm}$. This aspect has a significant benefit for the spectral analysis: as water is omnipresent, there is always a clear absorption feature that we can use for exact frequency determination, providing a robust anchor for frequency locking by using its spectral position as a control parameter for an active feedback loop on the laser heat sink.

## 2.2 Instrumental design

The instrumental design is strongly driven by the targeted application with special focus on compactness, ruggedness, low power consumption and weight. These aspects support an open-path approach, where the absorption cell is directly exposed to the atmosphere at ambient pressure, eliminating the need for gas handling and a sampling pump. This considerably reduces weight and power consumption, leads to fast response, and assures low contamination. For high-precision trace gas detection, however, it is necessary to use a multipass cell (MPC) that folds the light path to achieve a sufficiently large signal-to-noise ratio

(SNR). This key element often limits miniaturization of laser spectrometers, mainly because of its bulky design. Therefore, we fundamentally reconsidered the MPC concept and recently developed a novel and versatile solution, which is unique in its combination of size, mechanical rigidity and optical stability. A full description of this design, called segmented circular

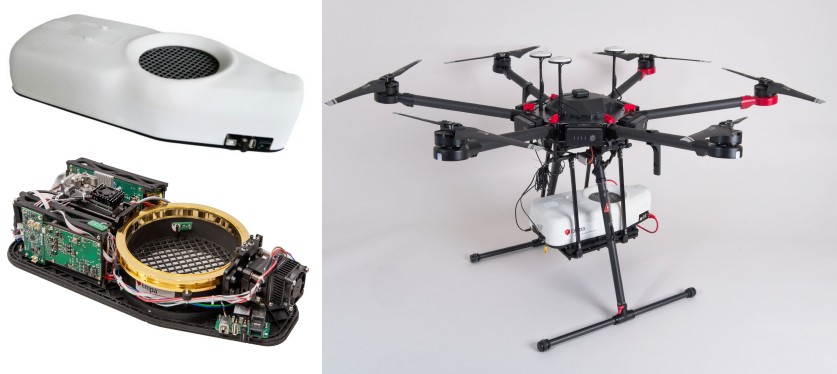

**Figure 2.** Photograph of the $CH_4$ laser spectrometer with and without the protecting cover (left). The main building blocks are the TO3-packaged QCL installed on a three-axis stage with additional TEC stabilized heat sink, the SC-MPC with $10\,m$ effective path length, the MCT detector, also with TEC stabilized heat sink, and five PCBs incorporating the hardware electronics: an icw laser driver, a DAQ unit with extension board, a power-supply module, the OEM TEC-board of the detector, and a custom TEC-board for laser and detector heat sinks. The heat sinks of the laser and detector are, in addition, actively air cooled with fans. Picture of the $CH_4$ sensor installed on the hexacopter (Matrice 600, DJI).

multipass cell (SC-MPC), is given by Graf et al. (2018). Here, we only recall the main characteristics of the SC-MPC. It consists of a monolithic aluminum ring with a radius of $77.25\,mm$, containing sixty-five quadratic, spherically curved segments

seamlessly shaped into the ring's inner surface. This geometry supports up to $10\,m$ optical path length at a total mass of less than $200\,g$. The SC-MPC has a confocal configuration, and it can thus directly accept the collimated laser beam emitted by the DFB-QCL (Alpes Lasers, Switzerland) encapsulated into a TO-3 housing with embedded collimating optics and thermoelectric cooler (TEC). Furthermore, the cell is largely insensitive to the aiming and coupling of the laser beam, which allows using a sturdy, custom-built three-axis linear stage for the laser source assembly. This module contains an additional TEC connected

to a heat sink, which is cooled by forced convection using a fan. After undergoing 65 reflection, the light is focused onto the IR-detector by the specially shaped last segment of the MPC cell. The detector module (Vigo System, Poland) consisting of a multi-junction HgCdTe photodiode (PVM-2TE-8-1$\times$1) with a DC-coupled preamplifier (SIP-DC-15M-TO8-G) is attached to the opposite side of the ring cell (see Fig. 2). The thermal management of the detector is implemented in the same fashion as for the laser, because of the elevated sensitivity of the detector signal to its heat sink temperature (see below). The thermal

stability of the detector is crucial for low-drift operation. A 2-inch solid-Ge etalon, mounted on a periscope-like mechanical support, can be placed into the MPC to determine the laser tuning-rate. The opto-mechanical setup is thus realized without any beam steering and shaping optics and has an overall weight of $750\,g$.

Of similar importance are the controlling and driving hardware as well as the data acquisition system. Here, we adopted our intermittent continuous wave (icw) laser driving concept (Fischer et al., 2014) and developed a current-source based laser

driver optimized for a small footprint and a high temperature stability. The design relies on the concept recently published by

Liu et al. (2018), and features three main elements: i) an active current control, provided by an operational-amplifier based current regulator, ii) a high-precision analog pulse generator to generate current pulses with an adjustable amplitude, shape, and width; and iii) a capacitor that acts as a low-noise power supply, being disconnected from the external source during the operation of the QCL. The whole circuit was optimized for low noise and low temperature sensitivity.

The laser was driven by current pulses of $88\,\mu s$ duration at $55\,\%$ duty cycle, corresponding to 6000 scans per second. Hardware control and DAQ functions are provided by an integrated System-on-Chip (SoC, STEMlab 125-14, Red Pitaya, Slovenia) open-source platform featuring an integrated dual-core ARM Cortex A9 processor with Xilinx 7-series FPGA logic. The FPGA firmware (VHDL-Code) and the Linux service routine (C-Code) were custom developed. The SoC GPIOs provide digital triggers required for operating the QCL driver. Further functionalities are realized by a custom-developed extension board with additional ADCs (16- and 24-bits) and DACs. These are used to set and read several temperature values for spectral and diagnostic purposes. The hardware internal communication is based on the Inter-Integrated Circuit ($I^2C$) protocol. The temperature control of the QCL relies on a high precision thermoelectric cooler (TEC) controller (WTC3243, Wavelength Electronics, USA), whilst the additional heat sinks (laser and detector) are stabilized by a custom-developed PWM based controller. The FPGA contains a state-machine that is clocked eight times slower than the sample clock ($125\,MS/s$), which makes routing less critical, i.e. larger data path delays are tolerated. For higher flexibility in data acquisition, several user defined time-windows within a spectral scan are supported. By means of decimation (up to 8), the total acquisition time can be increased, while the surplus samples can be dropped or averaged. The summation of consecutive spectra is implemented as DSP-Adders and Dual-Port-BRAMs. Port $A$ of the BRAMs is for storing the new sum of the spectra, while port $B$ is for reading the last sum. The spectra are then transferred from the programming logic (PL) via AXI-FIFO, DMA and DDR-RAM to the processing system (PS). The DMA is configured from a thread started by the Linux service routine. This thread polls the IRQ-bit of the DMA status and retriggers the DMA after each interrupt to copy the spectrum to a new destination address within a ring buffer. Whenever a client requests a spectrum, the current data is read from the ring buffer and sent to the client via TCP/IP. Two independent instances of the Linux service routine are started with their own TCP/IP port to assure that the writing and reading of additional signals (e.g. analog and digital inputs) do not interfere with the fast query of spectra.

The detector signal is digitized by the RP's on-board ADC with 14-bit resolution and a sampling rate of up to $125\,MS/s$, while the FPGA performs the real-time and on-board averaging of the data-stream as described above. These data, together with environmental and system log-data, are continuously saved on a USB flash-memory and, optionally, a copy is sent to a host-PC via TCP/IP using wired or wireless connection. Environmental parameters, such as barometric pressure, relative humidity, and ambient temperature are provided by an integrated PHT combination sensor (MS8607, Measurement Specialties Inc., USA) situated at the lower edge of the multipass cell (see Fig. 2). Finally, a GPS module (Adafruit Industries, USA) for altitude and position retrieval rounds-up the fully autonomous sensor, capable of delivering all the necessary parameters for mobile laser spectroscopy applications. The modular elements are fixed on a mechanically rugged 3D-printed polycarbonate platform with an embedded battery case. A lightweight cover assures protection and insulation of the device (Fig. 2). The overall instrument weights $2.1\,kg$ including the LiPo battery ($18\,V$) and has an outdoor average electrical power consumption of $18\,W$.

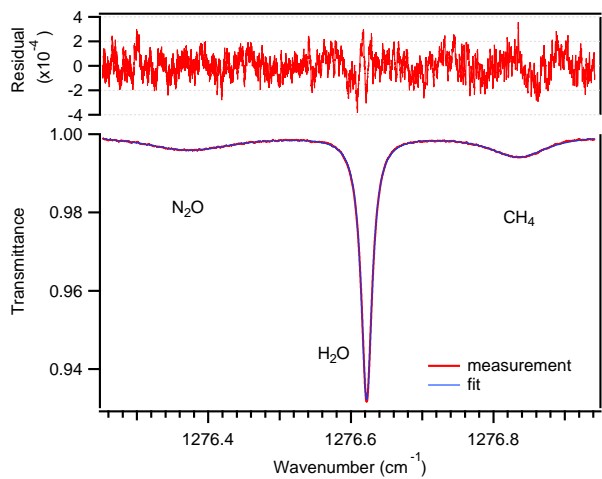

**Figure 3.** Measured transmission spectrum of ambient air at atmospheric pressure and room temperature recorded at $1\,Hz$, i.e., averaged over 6000 consecutive single scans. The associated fit (Voigt profile) and the residual plot (top) indicate the high fidelity and quality of the measurement. The laser frequency tuning can be extended to cover a nearby absorption line of $N_2O$ as well.

For the flight deployments, the device is mounted beneath a hexacopter (Matrice 600, DJI, China) on a bay plate using a gimbal frame with anti-vibration rubber dampers (see Fig. 2). The maximum flight time with the $2\,kg$ payload is about $20\,min$. During this period, a wireless bi-directional data link (SkyHopper PRO, Israel) assures real-time access to the raw spectral data and all hardware-related parameters. This allows for real-time spectral fitting and logging, and the user has full control over the entire hardware settings and can continuously monitor status of the instrument.

The spectral analysis takes place on the averaged data of 6000 spectra corresponding to $1\,Hz$ time resolution. A typical transmission spectrum with the associated fitted curves using Voigt profiles is shown in Fig. 3. The spectral line intensity and the broadening parameters are taken from the HITRAN2016 database (Gordon et al., 2017), whereas the gas pressure and temperature are measured.

## 3 Results and Discussion

### 3.1 Characterisation and Validation

The controlled characterization of any open-path spectrometer under representative flight conditions is highly challenging. Therefore, we custom-built a small ($60\,l$ volume) climate-chamber to have full control over the parameters of interest, such as ambient pressure, temperature and methane mole fraction (given in units of parts-per-million or billion (ppm or ppb) throughout this paper, representing the number of moles of methane per moles of air). Equipped with a high-power TEC-assembly (with a total of $250\,W$ cooling power) mounted on a water-cooled heat sink (Lytron Inc., USA), the chamber allows for rapid temperature modulation. A turbulent air circulation can be realized by operating a large radial fan with a flow-rate of $55\,m^3\,h^{-1}$.

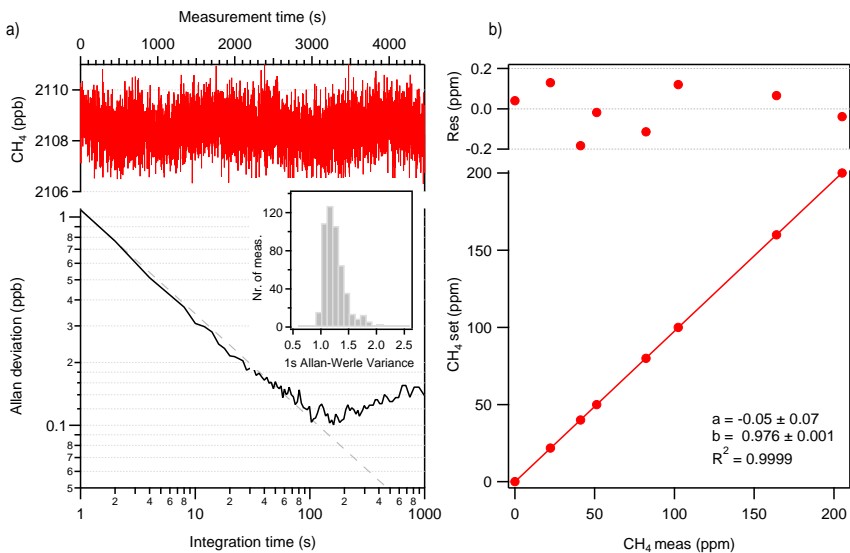

**Figure 4.** a) Allan-Werle deviation plot as a function of integration time. The intercept at $1\,\mathrm{s}$ corresponds to $1.1\,\mathrm{ppb}$ precision, which continues to improve for $100\,\mathrm{s}$ reaching a value of $0.1\,\mathrm{ppb}$. The inset is a histogram plot showing the distribution of the Allan deviation values at $1\,\mathrm{s}$ over multiple measurements. b) Calibration and linearity test of the $CH_4$ sensor.

The precision and stability of the spectrometer was assessed within the chamber under continuous purging with air at $2\,\mathrm{l\,min^{-1}}$ from a pressurized air cylinder. During this test, the pressure and the temperature were maintained constant, while the fan was running at full speed. Figure 4a summarizes the results of this measurement. The Allan-Werle variance technique (Werle et al., 1993) applied to the data collected over one hour shows that the device achieves a precision of $1.1\,\mathrm{ppb}$ within one second, corresponding to an absorption noise level of $5.8 \times 10^{-6}$. The sensitivity can be further improved by averaging over at least another $100\,\mathrm{s}$. After this time, the two-sample variance, deviates from the expected $1/\tau$ line, which indicates that drifts start to dominate the system. Nevertheless, the measurement precision stays bellow $0.5\,\mathrm{pbb}$ for $20\,\mathrm{min}$, corresponding to the duration of a common drone flight. Repeated precision tests indicate a robust stability of the system. The inset in Fig. 4 shows the distribution of the $1\,\mathrm{s}$ Allan-Werle deviations (over 500 values) calculated from data within $200\,\mathrm{s}$ time slots. These records were taken at various days, operating the instrument in the laboratory in open-path configuration and measuring ambient air. The histogram is slightly skewed to the right with the highest count located at $1.2\,\mathrm{ppb}$. It shows that the $1\,\mathrm{s}$ noise level is highly reproducible and adds confidence that the $1.5\,\mathrm{h}$ Allan-Werle deviation plot shown in Fig. 4 is representative for the performance of the instrument.

The chamber was also used to determine the response of the spectrometer to $CH_4$ concentration changes. For this purpose, a certified calibration gas with high $CH_4$ concentration ($200\,\mathrm{ppm} \pm 1\,\%$, PanGas, Switzerland) was dynamically diluted with dry nitrogen ($N_2$) in a step-wise fashion using mass flow controllers (see Fig. 4b). The derived linear calibration slope was then used throughout the field campaigns.

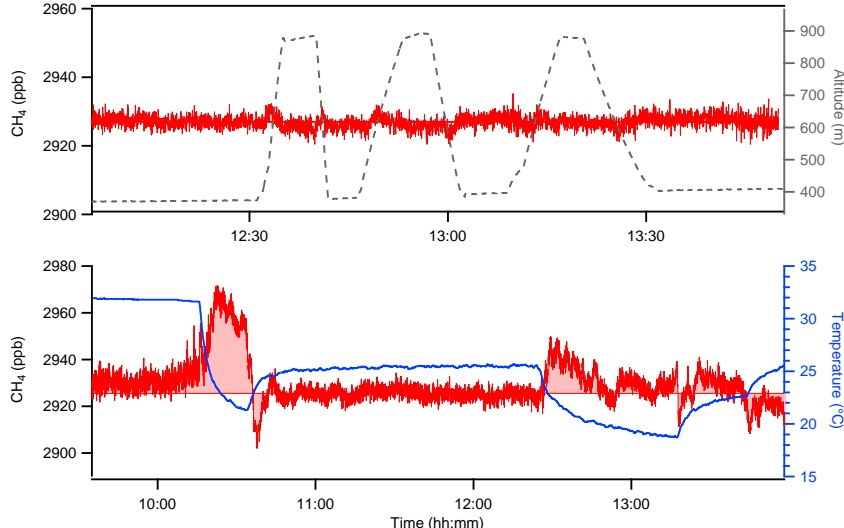

**Figure 5.** Pressure and temperature dependency of the retrieved $CH_4$ mole fraction values. The induced changes in pressure and temperature correspond to the variations expected during flight, mimicking changes in altitude up to $500\,\mathrm{m}$ and related temperature gradients.

A critical and limiting factor for trace gas laser spectroscopy is the appearance of unwanted etalon effects, generally referred to as fringing. This effect is induced by multiple reflections from partially reflecting plane-parallel surfaces (windows, focusing optics) interfering with the main beam. As they differ in phase due to the different path lengths traversed, they can efficiently translate optical frequency into transmitted intensity. This inherent transfer function may exhibit temporal variations due to mechanical vibrations or changes in ambient temperature or pressure. This is especially critical under field conditions, i.e. during flight of the spectrometer, as it tends to impair the measurement. To investigate the impact of such influences, the instrument was repeatedly exposed to sudden pressure and temperature variations generated in the climate chamber. The span of these changes corresponds to situations expected during realistic flight scenarios, e.g., assuming variations in flight altitude up to $500\,\mathrm{m}$ relative to ground, related temperature gradients as well as diurnal temperature changes of up to $10\,°\mathrm{C}$. The results of these experiments are summarized in Fig. 5.

It was found that pressure variations do not influence the performance, as the real-time on-board pressure measurements can be used to fully account for the respective impact on the spectral data, i.e. changes in absorption line width (pressure broadening) as well as changes in the number density of molecules (ideal gas law). In contrast, variations in temperature had a significant effect on the retrieved $CH_4$ mole fraction. This is mainly attributed to changes in the opto-mechanical system. In particular, we found that changes in the retrieved $CH_4$ mole fraction were linearly correlated ($23\,\mathrm{ppb\,K^{-1}}$) with the temperature of the detector module. Therefore, we implemented a TEC-based solution to maintain its heat sink temperature at a constant value of $32\pm0.05\,°\mathrm{C}$ even under varying environmental conditions. Further influencing factor, but less significant in magnitude, was the effect of dynamic fringe structures. The broad line-profile of the $CH_4$ absorption line was susceptible to slight shifts in the laser emission frequency, which caused subtle baseline variations. Given their complex behaviour, it is not possible to fully

account for these fringes, which, therefore, may bias the retrieved concentrations. Our strategy to minimize their impact was to implement an active frequency locking scheme using the water absorption line position as a reference. Thereby, the peak position is continuously determined by real-time spectral fitting and compared to the initial value. Whenever the difference is larger than $\pm 2 \times 10^{-4}\,\mathrm{cm}^{-1}$, the laser heat sink temperature is repeatedly adjusted in small steps (mK) with a latency time of $10\,\mathrm{s}$ to compensate this drift. By implementing all the above optimisation strategies, the effect of sudden temperature changes on the retrieved $CH_4$ concentration was reduced to about $4\,\mathrm{ppb\,K^{-1}}$, as indicated on Fig. 5. These perturbations are rather of random nature, i.e. difficult to account for, and most likely reflect the susceptibility of the entire electronics to abrupt temperature changes. Nevertheless, considering the typical flight scenarios (e.g. emission screening), the temperature changes in such cases are only a few degrees, and correspondingly, the instabilities in the $CH_4$ concentration are expected to stay bellow $10\,\mathrm{ppb}$. Nevertheless, for high-altitude flights, such as PBL determination, it is definitely advisable to adopt a more elaborated thermal stabilization scheme. In principle, there are two ways to further improve the temperature stability of the instrument: active (e.g. TEC-based) and passive. While the former can be very compact, it requires significant electrical power. A passive isolation although can efficiently dampen sudden temperature fluctuations, it has limited flexibility for the range of supported temperatures. Thus, during cold season a good isolation can be beneficial, while on hot summer days can lead to thermal rollover. Lately, we were successfully applied a combination of TEC and phase-change material (as thermal buffer medium) solution, but with considerable cost of weight and electrical power (Graf et al., 2020). In this case, the instrument demonstrated temperature stabilization in a range of nearly $80\,^{\circ}\mathrm{C}$. Similar approach could be implemented also for $CH_4$ spectrometer, when the target application requires the highest precision and accuracy.

Finally, we also tested the influence of mechanical vibrations and air turbulences generated by an operating drone. For this purpose, the instrument was installed on the hexacopter, which was then turned on and off repeatedly to generate typical vibrations and turbulence. It became immediately clear that a direct mechanical contact with the drone-body propagates the rotor vibrations onto the spectrometer, generating significant optical noise. Therefore, a gimbal frame with anti-vibration rubber dampers was added beneath the hexacopter and the $CH_4$ spectrometer was then firmly mounted on this platform. Followup measurements showed no effect on the retrieved methane mole fractions within the measurement precision and natural variability, proving the efficiency of the gimbal system.

## 3.2 Field deployments

Prior to flying aboard a drone, the spectrometer was installed and operated continuously outside in the vicinity of a monitoring station of the Swiss National Air Pollution Monitoring Network (NABEL) in Dübendorf as a final verification for field endurance and long-term stability. Due to the open-path configuration, the measurements were only conducted under good weather conditions to avoid exposure to precipitation. Nevertheless, the instrument was experiencing large diurnal temperature variations (up to $10\,^{\circ}\mathrm{C}$ difference, see top-plot on Figure 6), wind, and occasionally also direct sunlight. Simultaneously, regular $CH_4$ monitoring measurements were performed with a cavity ring-down spectrometer (G1301, Picarro Inc., USA) installed at the NABEL station. The lateral distance between the air-inlet and our $CH_4$ sensor was about $1.5\,\mathrm{m}$. An example of daily $CH_4$ time-series is shown in Fig. 6. The QCLAS-device demonstrated stable operation over extended periods of time and suc-

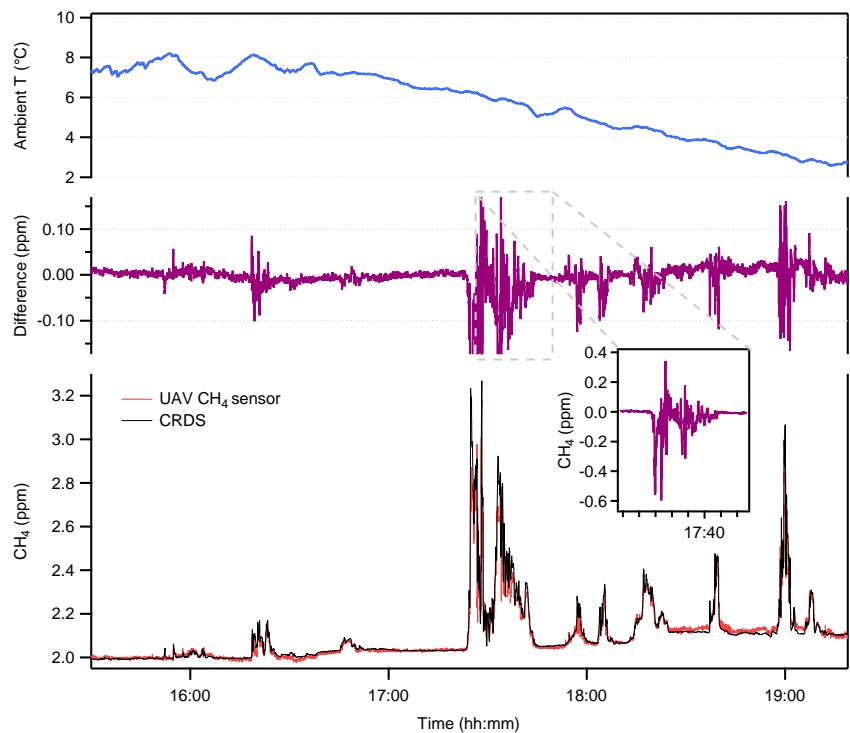

**Figure 6.** Intercomparison of the $CH_4$ spectrometer with a reference methane analyzer at the NABEL monitoring station. The device was placed outside, fully exposed to changing environmental conditions, in the proximity of the air inlet located at $2\,m$ above ground. The recorded time-series and the difference in the $CH_4$ concentration between the two analyzers are shown in the bottom and middle plot. The period of large variation in the atmospheric methane concentration is detailed in the inset figure. The variation of the ambient temperature during the measurements is given in the top graph.

cessfully captured sudden concentration variations in atmospheric methane. The observed spikes were reoccurring regularly in the evening hours, but their origin/source is unknown to us. However, they offered a good opportunity to assess the dynamic response and sensitivity of our $CH_4$ instrument. The observed differences between the two analyzers (see middle plot in Fig. 6) are mainly due to the unsynchronized clock's and the difference in response times as well as in sampling strategies, i.e. open- and closed-path.

### 3.2.1 Flying above an artificial source

For the first test flight we used a small grassland area located nearby the Empa campus. Here, we installed an artificial $CH_4$ source and released pure methane at a constant flow-rate of $10\,l\,min^{-1}$. To simulate the concentration field of methane and calculate the footprint emission map, the GRAL dispersion model (Oettl et al., 2002) was used. This model computes high-resolution wind fields and turbulence in the presence of 3D obstacles (buildings and vegetation) using a computational fluid

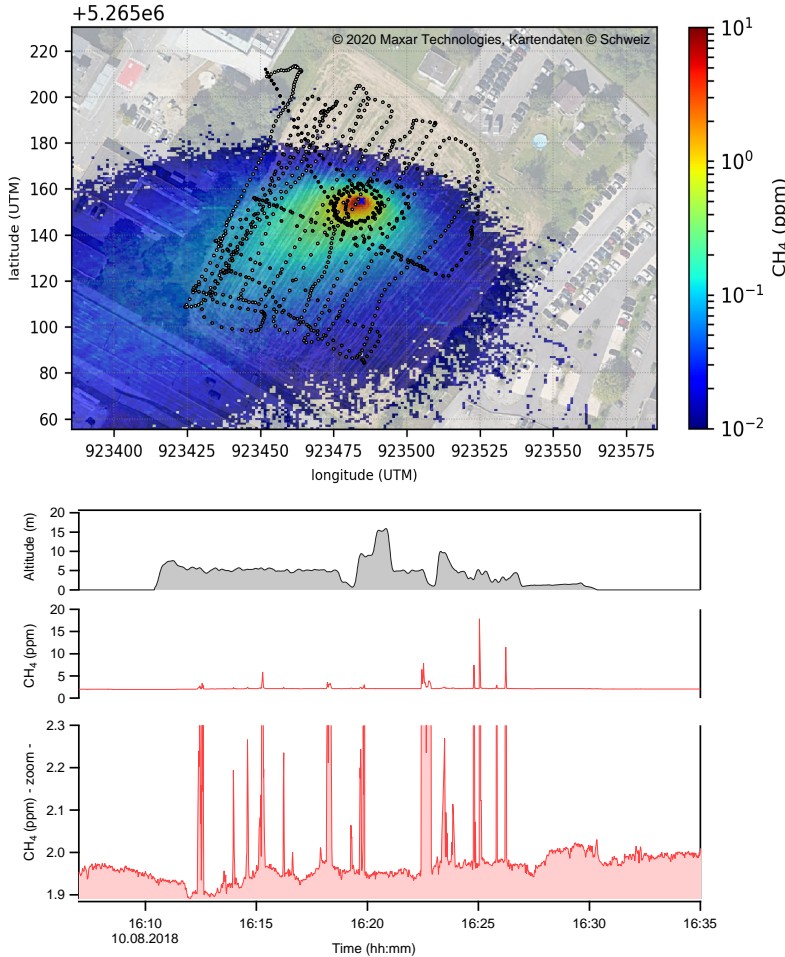

**Figure 7.** Satellite image of the area used for controlled surface release with the artificial $CH_4$ source, indicated by the blue cross, with the flight tracks superimposed on the simulated footprint emission map (top). Time-series of the retrieved $CH_4$ concentrations with a zoom-in around the background level (bottom). The in-flight period can be easily deduced from the altitude plot (bottom).

dynamics (CFD) approach. The dispersion of pollutants is then computed in a Lagrangian framework by releasing virtual particles at prescribed sources (Anfossi et al., 2006). The forcing for the simulation is obtained from flow fields computed by averaging locally observed meteorological data. In this simulation, meteorological data during the flight were obtained by placing a 3D sonic anemometer (uSonic-3 Wind Sensor, METEK GmbH, Germany) in the middle of the field. The anemometer, mounted on a $5\,\mathrm{m}$ high mast, sampled the wind direction, ambient temperature, and wind speeds in the $x$, $y$, and $z$ directions at

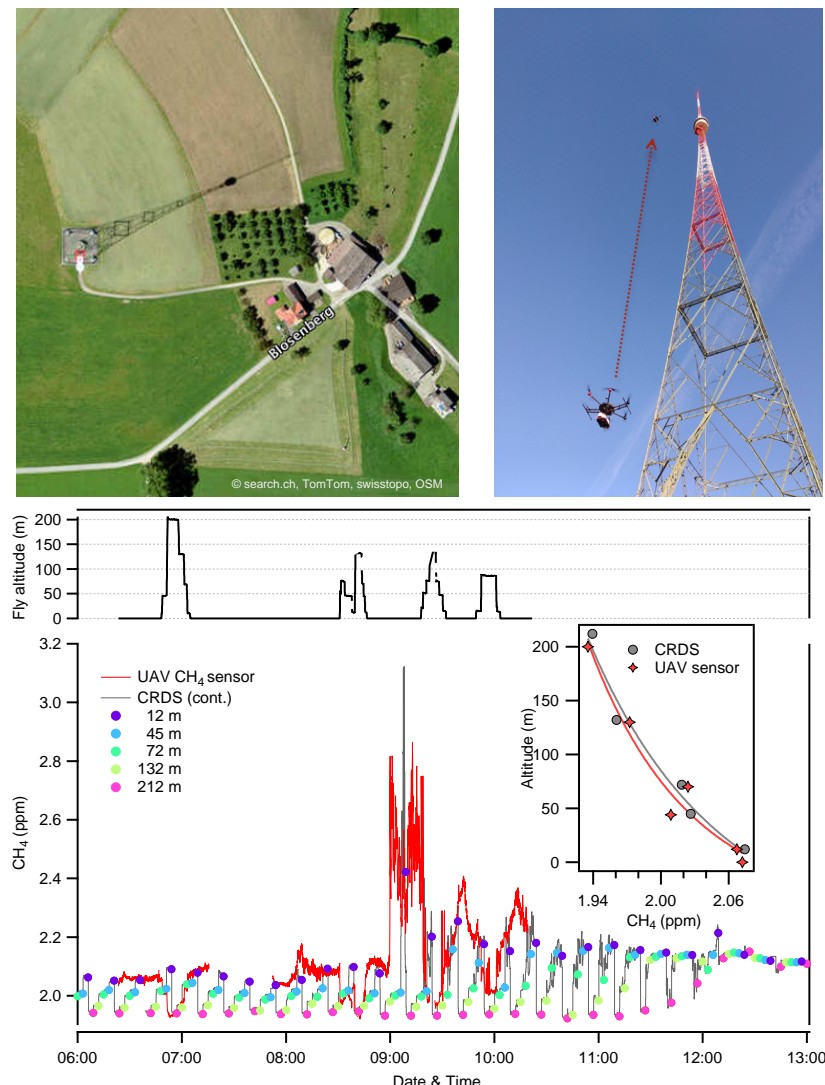

**Figure 8.** Satellite image of the area used for profiling (top left). Photograph of the drone flying various heights in the vicinity the radio tower (212 m) at Beromünster (top right). Vertical profile measurements of atmospheric methane (bottom). The concentration gradient is clearly observable until noon, when the nocturnal boundary layer is broken up by convective vertical mixing. The gray trace represents the continuous, while the colored dots indicate the mean $CH_4$ concentration values at various heights as measured by a ground-based CRDS analyzer. The UAV methane instrument (red trace) performed four flights covering the altitude span between 0 to 200 m. The inset shows the comparison of the profile data collected by the two analysers. The concentration gradient is well captured in both cases.

50 Hz. Meteorological data from the anemometer were then averaged to regular $5\,min$ intervals to obtain turbulent parameters such as Obukhov length, friction velocity, and wind speed fluctuations ($u'$, $v'$, and $w'$), and served as input to compute the flow fields. Simulated concentration field of methane at $5\,min$ intervals were finally averaged to obtain a footprint emission map of the whole modelling domain.

The $CH_4$ spectrometer was operated prior the flight for 20 minutes on the ground, including warm-up time, recording an etalon spectrum, setting up the spectral fit, initiating the data recording and the wireless communication. Afterwards, the drone was flown at heights above ground ranging from 2 to $15\,m$ for about $20\,min$ in the proximity of the artificial methane source (see Figure 7). The time-series recorded along the trajectory intersecting the plume exhibit large spikes reflecting methane enhancements from the release, which is in good qualitative agreement with the GRAL model, which predicted methane concentrations enhancements of up to $10\,ppm$ at $5\,m$ above the ground.

### 3.2.2 Atmospheric profiling

A challenging but important application for drone-based $CH_4$-sensors is to interrogate the atmospheric boundary layer (ABL) frequently and at multiple locations to provide feedback for ABL model developments. Therefore, we conducted a field test to assess the capability of our $CH_4$ spectrometer to detect small, natural inhomogeneities in the atmospheric methane concentrations at different altitudes throughout the ABL. These field experiments were conducted at Beromünster, which is located in a moderately hilly environment at the southern border of the Swiss Plateau. Here, a former radio tower ($47° 11' 23'' N$, $8° 10' 32'' E$, $212.5\,m$ tall, base at $797\,m$ a.s.l.) is equipped with air inlets at 12.5, 44.6, 71.5, 131.6 and $212.5\,m$, respectively (see Fig. 8). A cavity ring down spectrometer (G2401, Picarro Inc., USA) monitors the mole fractions of $CH_4$ at these five different heights sequentially, sampling each height for $3\,min$ (Satar et al., 2016). The field-test took place on a day with stable atmospheric conditions, when methane accumulates during nighttime in a shallow nocturnal boundary layer near the ground. Figure 8 shows the $CH_4$ time series provided by the CRDS analyzer for the selected day. Consistent with our expectations, the data indicate a pronounced gradient among the inlet heights during the time-period before sunrise. This concentration build-up is then slowly disappearing after sunrise due to break-up of the nocturnal boundary layer by convective vertical mixing. The sensor's precision was adequate to easily resolve the vertical gradients of atmospheric methane. The $CH_4$ mole fractions typically varied between 1940 and $2200\,ppb$, with some exceptional cases, when sudden increase of $CH_4$ mole fractions of up to $3200\,ppb$ were observed close to ground over time periods of tens of minutes. These short-term spikes were attributed to pollution events, due to emissions from the farmsteads (ruminants) in the close vicinity ($200\,m$) of the tower.

### 4 Conclusions

A compact and lightweight laser spectrometer capable of atmospheric $CH_4$ measurements at high sensitivity and time resolution ($1\,Hz$) was developed to be deployed aboard UAVs. The open-path system has an overall weight of $2.1\,kg$ (including battery) and an electrical power consumption of $18\,W$. It can easily be carried by commercial drones, providing unique opportunities for mobile sensing and spatial mapping of $CH_4$ mole fractions at unprecedented precision. The fully autonomous

device uses a QC laser emitting in the mid-infrared at $7.83\,\mu\text{m}$, a versatile and robust segmented circular mirror multipass cell (SC-MPC) with $10\,\text{m}$ effective optical path length, and a custom-developed low-power SoC FPGA based data acquisition system. The instrument was characterized in the laboratory in terms of precision, linearity, and its dependence on various environmental factors, and then validated under field conditions and flown in two different application scenarios. With a precision of $1\,\text{ppbv}\ CH_4$ at $1\,\text{Hz}$, the spectrometer represents the very first portable instrument of its class regarding precision, weight, and real-time analysis capabilities. It outperforms existing solutions based on near- and mid-infrared spectroscopy using CRDS (Berman et al., 2012; Martinez et al., 2020) and wavelength modulation (Golston et al., 2017) techniques by at least an order of magnitude. The large dynamic range and the fast time resolution of the open-path system can be used to capture transient $CH_4$ concentrations and thus localize and quantify a wide variety of emission sources. Furthermore, the high precision enables measuring vertical profiles of methane, thus creating new opportunities for the study of trace gases in the atmospheric boundary layer, but also for investigating diffuse natural methane sources. By the time of writing, the device has been deployed in over 60 flights as part of a large field-campaign (ROMEO within MEMO[2] project) measuring emissions from the oil- and natural gas infrastructure facilities in Romania.

*Author contributions.* B.T. and L.E. conceptualized the project. M.G. designed and developed the SC-MPC, implemented the environmental sensor and GPS communication scripts on the SoC, and constructed the climate chamber. P.S. developed and realized the hardware electronics. A.K. developed the FPGA functionalities for real-time spectral averaging and on-board data saving. H.L. designed and developed the hardware control and data processing software. J.R. contributed to the laboratory and field experiments, performed calibration and validation measurements, and flew the drone. R.P.M. performed the simulation of the concentration field of methane and calculated the footprint emission map using the GRAL dispersion model. B.T. designed, set up and optimized the spectrometer, coordinated and performed laboratory and field experiments, analyzed the data, and prepared the manuscript with contributions from all authors. L.E. supervised the project, discussed the results, and reviewed the paper.

*Data availability.* The data used in this manuscript are available from the corresponding author upon request.

*Competing interests.* The authors declare that they have no conflict of interest.

*Acknowledgements.* The authors would like to thank all those who have contributed to the $CH_4$ sensor development (Chang Liu, Badrudin Stanicki, Curdin Flepp, Nico Schäfer, Sebastian Humbel) and field testing (Killian Brennan). We thank Beat Schwarzenbach for the CRDS data and technical support during the field measurements at the NABEL station in Dübendorf. Markus Leuenberger from University of Bern (Switzerland) is acknowledged for making the CRDS data from Beromünster available. This research was partially founded by ABB Switzerland and we specially thank the contribution of Deran Maas. Further founding was provided by MEMO[2], a European Training

Network (MSCA-ETN). M. Graf was supported by the Swiss National Science Foundation (SNSF) under grant no. 157208. We thank Alan Fried and Dave Nelson for their kind reviews and helpful suggestions for improving the manuscript.

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
