# Peer review of "A compact QCL spectrometer for mobile, high-precision methane sensing aboard drones"

_Atmospheric Measurement Techniques, 2020_

## Referee Comment (RC1) · Alan Fried (Referee) · 9 May 2020

This is an excellent paper and a real joy to read. The authors did a great job in both discussing their new instrument and in validating the performance in a variety of test environments. Such attention to detail is commendable and is not typically found in such new developments.

This paper is acceptable for publication with only 4 very minor points that this reviewer would like to see addressed.

1. It would be very informative to the reader if the authors could indicate how reproducible the Allan-Werle results of Fig. 4a were obtained. Does this figure represent typical performance or does this represent the best series of measurements? For that

purpose it would be very interesting to plot a histogram of the 1second Allen-Werle deviations if the authors indeed recorded multiple plots.

2. Regarding the temperature sensitivity of their instrument, is there any possibility to further stabilize the temperature of the electronics and/or the entire optical system either actively or passively employing better insulation? Although the 4 ppb/K sensitivity (not 4 ppb/K-1) is quite good, 10 degree C temperature changes, as would be experienced by changing altitudes, seem to affect performance for time periods $\sim$ 20 - 30 minutes (Fig. 5). It would be nice to mitigate this long equilibration time period.

3. It would be useful to indicate the $H_2O$ sensitivity of their retrieved $CH_4$ results since situations where the $H_2O$ mixing ratios can approach up to 3-4 times the 1% levels simulated.

4. The authors may wish to explain the slight UAV overestimate of $CH_4$ relative to the CRDS in Fig. 8 at just after 08:00 at 12 m sampling height.

———————————————————

---

## Short Comment (SC1) · 13 May 2020

Hi, First of all, thank you for your article which I found very interesting and of great quality. I think that some details and figures about the acquired spectra would be very interesting. It seems that the experimental spectra showed in figure 2 corresponds to the region of spectral interest but was it obtained using your instrument ?

Did you scan the same spectral range during during the UAV flights ? In the presented spectra of figure 2, it seems that you manage to fit the $N_2O$, $CH_4$ and $H_2O$ absorption lines quite nicely, so why don't you also present $N_2O$ and $H_2O$ concentration in your results ? Since the two Picarro models (G1301,G2401) used in your field experiment are able to analyse $H_2O$, I think that it would be very interesting to share $H_2O$

concentration results, or even a comparison with Picarro sensors.

A 1Hz raw spectra obtained during a flight would also be interesting to show as you mention baseline variations and dynamic fringes structures

As mentioned line 209, prior to the flight, an etalon spectrum is recorded and spectra fit is set up; is it part of your standard operation protocol ? Do you always use this procedure in order to compensate an potential drift from the QCL over time ?

Thanks again for sharing this work

Best regards,

N.Dumelié

---

## Referee Comment (RC2) · David Nelson (Referee) · 17 Jun 2020

This is an excellent paper describing a truly impressive accomplishment: the dramatic miniaturization of a laser based trace gas monitor with very little sacrifice to measurement accuracy. The results are convincing and the paper is very well written and should certainly be published. I have a few small suggestions for the authors to consider but the suggestions are not mandatory.

1) Taking advantage of the surprisingly narrow water line is very clever. However, the discussion describing why this line is narrow is not clear. Does it simply have a small broadening coefficient or is it narrow because of Dicke narrowing which does not, I think, involve energy level spacings. Or is it both? It would be nice to clarify this

discussion or remove it if the explanation is not clear.

2) It is a little surprising to me that the detector temperature has a strong effect on the reported mixing ratio. Do the authors have an explanation for this? Changes in linearity or bandwidth, perhaps?

3) The statement near the end of Section 3.1 that the remaining temperature artifacts "most likely reflect the susceptibility of the entire electronics to abrupt temperature changes" seems unsupported. Do the authors have a reason for believing that the problem could not be due to an optical effect? If so that should be stated. If not, then perhaps it would be better not to speculate.

4) Figure 6 shows a comparison between the compact QCL monitor and a Picarro monitor. The scale is so large that it is difficult to see the discrepancies. It would be useful to add a trace that shows the difference between the mixing ratios reported by the two instruments.

Again, this is an excellent paper and a great accomplishment!

---

## Author Comment (AC1) · 3 Jul 2020

**Authors' Response to Referee Comments**

We appreciate the overall positive response of the Referee and we would like to thank for his constructive comments and helpful suggestions on the manuscript, which helped us to further improve the clarity of the paper. Below, we give detailed responses (in blue) where appropriate.

Alan Fried (Referee)

This paper is acceptable for publication with only 4 very minor points that this reviewer would like to see addressed.

1. It would be very informative to the reader if the authors could indicate how reproducible the Allan-Werle results of Fig. 4a were obtained. Does this figure represent typical performance or does this represent the best series of measurements? For that purpose, it would be very interesting to plot a histogram of the 1 second Allan-Werle deviations if the authors indeed recorded multiple plots.

We agree that is tempting to pick the most attractive time-series sequence to show the instrument performance. However, in our case, it is very difficult (as already mentioned in the manuscript) to maintain a constant concentration level, at the high precision level of the instrument, over prolonged period of time, because of the open-path configuration. Thus, the only well-controlled experiment took about 3 hours, from which 1.5 hours were spent for purging, and the other 1.5 hours are shown in Fig.4. For the 1 second Allan-Werle deviations, the long-term stability of the environment is not critical. Therefore, we analyzed our repeated overnight measurements, where the instruments simply measured laboratory air. We collected more than 30 hours of data and calculated the 1 s Allan-deviation of every 200 s period. The histogram plot of the corresponding 520 values is now is added to Fig.4a as an inset. It shows that the 1 s noise level is highly reproducible and adds confidence that the 1.5 hour Allan plot shown in Fig.4 is representative for the performance of the instrument.

2. Regarding the temperature sensitivity of their instrument, is there any possibility to further stabilize the temperature of the electronics and/or the entire optical system either actively or passively employing better insulation? Although the 4 ppb/K sensitivity (not 4 ppb/K-1) is quite good, 10 degree C temperature changes, as would be experienced by changing altitudes, seem to affect performance for time periods~20 - 30minutes (Fig. 5). It would be nice to mitigate this long equilibration time period.

Indeed, it is possible to develop additional solutions to achieve further thermal stabilization. In principle, there are two ways: active and passive. While the former can be very compact, it requires significant electrical power (e.g. TEC-based). A passive isolation can efficiently dampen sudden temperature fluctuations, but has limited flexibility for the range of supported temperatures. Thus, during cold season a good isolation can be beneficial, while on hot summer days can lead to thermal rollover. In our approach, we considered the most plausible applications (e.g. source identification and emission estimates), which require flight patterns that mainly involve close-to-surface surveillance or curtain-like profile flights. For these scenarios, the temperature fluctuations are less prominent. Nevertheless, for high-altitude flights, such as PBL determination, it is definitely advisable to adopt a more elaborated thermal stabilization scheme. Lately, we were successfully applied a combination of TEC and phase-change material (as thermal buffer medium) solution, but with considerable cost of weight and electrical power (see Graf *et al.*,

Compact and Lightweight Mid-IR Laser Spectrometer for Balloon-borne Water Vapor Measurements in the UTLS, Atmos. Meas. Tech. Discuss., https://doi.org/10.5194/amt-2020-243). This instrument demonstrates temperature stabilization in a range of nearly 80°C.

3. It would be useful to indicate the H2O sensitivity of their retrieved CH4 results since situations where the H2O mixing ratios can approach up to 3-4 times the 1% levels simulated.

We admit that having the $H_2O$ sensitivity on the retrieved $CH_4$ concentrations would complete the overall characterization of our device, however such investigation is not trivial (see also our reply to comment #1). Furthermore, we do not expect that during the flight (~20 min) the water vapor would drastically change in the atmosphere. Since we consider calibrating our QCLAS instrument prior flights with a ground-based CRDS instrument, either in-situ or taking bag samples that are analyzed in the laboratory afterwards, potential biases due to water vapor are minimal.

4. The authors may wish to explain the slight UAV overestimate of CH4 relative to the CRDS in Fig. 8 at just after 08:00 at 12 m sampling height.

There are in fact significant discrepancies between the two data set at 8 AM and in the following hours. We are, however, confident that these variations are real inhomogeneities of atmospheric methane concentration. As we mentioned in our manuscript, in the close vicinity (~200 m) of the radio tower is a farmsteads (ruminants), which has significant methane emission (see Fig.8 top left, the farm is located at the right). The drone flights were conducted on the small-unpaved road that connects the farm with the tower. Because of communication difficulties with the drone in the proximity of the metal frame of the radio-tower, the flights took place about 20 m from the tower, down on the roadside in the direction of the farm. Immediately after sunrise, we observed slight winds that unfortunately had east-west direction, i.e. we were measuring downwind to the farm. Therefore, the early morning data (before sunrise) is much more representative for the nocturnal boundary layer.

---

## Author Comment (AC2) · 3 Jul 2020

**Authors' Response to Referee Comments**

We appreciate the overall positive response of the Referee and we would like to thank for his constructive comments and helpful suggestions on the manuscript, which helped us to further improve the clarity of the paper. Below, we give detailed responses (in blue) where appropriate.

Dave Nelson (Referee)

This is an excellent paper describing a truly impressive accomplishment: the dramatic miniaturization of a laser based trace gas monitor with very little sacrifice to measurement accuracy. The results are convincing and the paper is very well written and should certainly be published. I have a few small suggestions for the authors to consider but the suggestions are not mandatory.

1. Taking advantage of the surprisingly narrow water line is very clever. However, the discussion describing why this line is narrow is not clear. Does it simply have a small broadening coefficient or is it narrow because of Dicke narrowing which does not, I think, involve energy level spacings. Or is it both? It would be nice to clarify this discussion or remove it if the explanation is not clear.

We are thankful for pointing out this weakness of the discussion. Indeed, the description was misleading and therefore we revised it to avoid any confusion. Just to clarify: The linewidth of an absorption at given pressure is determined by two different contributions, the Dicke narrowing and the pressure broadening. Pressure broadening is due to collisions, which perturb internal motion. Changes in rotational energy, especially for linear molecules, contribute sizably to linewidth. In general, broadening is small for high-$J$ transitions, because these states have energy separations of the order of 300 cm$^{-1}$ from other rotational states to which they can make collisional transitions. It is therefore difficult to absorb this energy in translational motion (kT ~200 cm$^{-1}$). (Eng *et al.*, Appl. Phys. Lett. (1972), 21, 303). On the other hand, a significant narrowing can be observed if (i) the quantum state of the molecule is not affected by collisions and (ii) the mean free path is smaller than the wavelength of the proving radiation. The high $J$-states have long rotational lifetime and thus exhibit also a significant collisional (Dicke) narrowing. This, however, manifests in the Doppler widths of spectral lines at higher pressures, which can be narrower than the width expected from the Maxwell-Boltzmann distribution (Giesen *et al.*, J. Mol. Spec. (1992), 153, 406-418). The collisionally narrowed line profiles, however, cannot be fitted to the Voigt function with the Doppler width expected from the Maxwell-Boltzmann distribution. The treatments of such line profiles requires more sophisticated profile functions that include the effect of soft- and hard-collision (e.g. Ngo *et al.*, Phil. Trans. R. Soc. A (2012), 370, 2495–2508). Since this effect is mainly dominating at gas pressures < 200 hPa, we decided to not mention its contribution in our paper, which is primarily focused at atmospheric pressure. Therefore, we modified the text in our manuscript correspondingly:

"*This can be explained by inefficient collisional relaxation due to the wide energy separation (~300 cm$^{-1}$) for high-J states of $H_2O$ (Eng et al., 1972; Giesen et al., 1992). Considering the typical energies of translation motion (kT ≈200 cm$^{-1}$), it is obviously difficult to take up the transition energies from such rotational states. Therefore, the perturbation of these high rotational energy states by collision remains low, which results in low pressure broadening, so that the $H_2O$ absorption line at ambient pressure appears as narrow as it would be at 0.1 atm.*

2. It is a little surprising to me that the detector temperature has a strong effect on the reported mixing ratio. Do the authors have an explanation for this? Changes in linearity or bandwidth, perhaps?

We share this surprise, but the observations strongly support this effect. Actually, we observed that mainly the shape of the signal profile was changing rather than the signal amplitude. We contacted the detector supplier and they pointed out that these small footprint preamp packages (SIP) have insufficient heatsinking capabilities, and it is mandatory to have additional, external heatsink to guarantee sufficient heat dissipation. We think that the Peltier-element used to cool the detector chip is overstrained, and that it loses temperature stabilization, resulting in the observed symptoms.

3. The statement near the end of Section 3.1 that the remaining temperature artifacts "most likely reflect the susceptibility of the entire electronics to abrupt temperature changes" seems unsupported. Do the authors have a reason for believing that the problem could not be due to an optical effect? If so that should be stated. If not, then perhaps it would be better not to speculate.

Yes, we have good reason. Since the optics is very compact, it is rather straightforward to induce temperature fluctuations affecting almost exclusively the optical elements only. Conversely, this is also true for the electronics. The "hot-spots" that we were able to identify are the potentiometers used to adjust the shape and amplitude of the current pulses for the detector. At less extent, a resistor element used in the reference current source can also be contributing to the instabilities/drifts of the laser driving current.

Although, these effects are minor, the broad linewidth of the methane absorption line is highly susceptible to any slight change in the laser emission. Currently, we are developing a newer version of our laser driver, where the potentiometers are replaced by digital signals.

4. Figure 6 shows a comparison between the compact QCL monitor and a Picarro monitor. The scale is so large that it is difficult to see the discrepancies. It would be useful to add a trace that shows the difference between the mixing ratios reported by the two instruments.

We added a plot to Fig.6 showing the difference between the mixing ratios reported by the two instruments.

---

## Author Comment (AC3) · 3 Jul 2020

**Authors' Response to the Short Comments**

We appreciate the comments. Below, we give detailed responses (in blue) where appropriate.

Nicolas DUMELIE (Short comment)

I think that some details and figures about the acquired spectra would be very interesting. It seems that the experimental spectra showed in figure 2 corresponds to the region of spectral interest but was it obtained using your instrument? Did you scan the same spectral range during during the UAV flights?

Yes, the spectra shown on Fig.2 was obtained with our mobile QCLAS instrument. Apparently, this point was not made clear enough. We slightly changed the text to improve this. For the flights, we were limiting the scanned region to $H_2O$ and $CH_4$ absorption lines, i.e. 1276.5 – 1277 cm$^{-1}$. This was mainly to optimize the acquisition time and improve the duty cycle of the measurements. Furthermore, we were aiming for a large section for the baseline to make the spectral fitting more robust.

In the presented spectra of figure 2, it seems that you manage to fit the N2O, CH4 and H2O absorption lines quite nicely, so why don't you also present N2O and H2O concentration in your results? Since the two Picarro models (G1301, G2401) used in your field experiment are able to analyse H2O, I think that it would be very interesting to share H2O concentration results, or even a comparison with Picarro sensors.

Our aim was to show that, in principle, the instrument has multi-species measurement capabilities, but our main target was the $CH_4$ concentration, for which we optimized the operation mode. For sake of completeness, it should be noted that, although the $N_2O$ looks nice, the precision is about 1 %, i.e. about 3 ppb. For many environmental applications, this might may not be good enough. However, if one considers artificial release experiments, where $N_2O$ is used as independent tracer, then this is attractive. The situation is similar for $H_2O$; we are using its absorption signal to precisely control the laser emission frequency and to easily identify the spectral region. Again, depending on the targeted application, measuring water vapor may be of interest, but for the time being we limited ourselves to $CH_4$.

A 1Hz raw spectra obtained during a flight would also be interesting to show as you mention baseline variations and dynamic fringes structures.

We consider that the best prove of the quality of spectra during flight is the time-series shown in Fig.7 (bottom plot). It is a zoom-in plot to illustrate the retrieved $CH_4$ concentrations during take-off, flight and landing. One can easily observe that there is no abrupt change in the time-series at the critical transitions. The baseline variations and dynamic fringe structures are very subtle and would be difficult to see without detailed analysis (see also our reply to Dave Nelson's comment #3).

As mentioned line 209, prior to the flight, an etalon spectrum is recorded and spectra fit is set up; is it part of your standard operation protocol? Do you always use this procedure in order to compensate an potential drift from the QCL over time?

Yes, this is our standard start-up protocol. The warm-up time of the instrument is about 20 minutes that one has to spend anyway. This time is well used to set up the fit, record a new etalon spectrum, establish communication with the host PC, mount the instrument on the drone, and make the drone ready for lift off. In our experience, the frequency drift of a state-of-the-art QCL is very small so that settings from a prior flight (that could be even weeks apart) can be readily used. In fact, we have now implemented an

automatic start-up procedure that takes care of all the initialization steps (except a new tuning curve measurement) and brings the instrument to full operation mode, without any input from the user.